# Increased risk of developing dental diseases in patients with primary Sjögren's syndrome— A secondary cohort analysis of population- based claims data

Chi-Jou Chuang[1,2], Chia-Wen Hsu[3], Ming-Chi Lu[2,4]*, Malcolm Koo[5,6]*

1 Division of Obstetrics and Gynecology, Dalin Tzu Chi Hospital, Buddhist Tzu Chi Medical Foundation, Chiayi, Taiwan, 2 School of Medicine, Tzu Chi University, Hualien, Taiwan, 3 Department of Medical Research, Dalin Tzu Chi Hospital, Buddhist Tzu Chi Medical Foundation, Chiayi, Taiwan, 4 Division of Allergy, Immunology and Rheumatology, Dalin Tzu Chi Hospital, Buddhist Tzu Chi Medical Foundation, Chiayi, Taiwan, 5 Department of Nursing, Tzu Chi University of Science and Technology, Hualien, Taiwan, 6 Dalla Lana School of Public Health, University of Toronto, Toronto, Ontario, Canada

* e360187@yahoo.com.tw (MCL); m.koo@utoronto.ca (MK)

**Data Availability Statement:** Due to legal restrictions imposed by the government of Taiwan in relation to the "Personal Information Protection Act", data cannot be made publicly available.

## Abstract

### Background

Although it is known that patients with primary Sjögren's syndrome (pSS) have impaired dental conditions, incidence rates and incidence rate ratios of various dental diseases in these patients are not clear. The aim of this study was to investigate the frequency and prevalence of dental diseases in patients with pSS, and to evaluate the risk of common dental diseases in these patients.

### Methods

A population-based retrospective cohort study was conducted using the data from the Taiwan's National Health Insurance Research Database. A total of 709 patients with newly diagnosed pSS between 2000 and 2012 were identified to form the pSS cohort. A comparison cohort of patients without pSS was assembled based on frequency matching for sex, 5-year age interval, and index year at a ratio of 10:1. All participants were followed until the end of the follow-up period or when the outcome of interest occurred. The incidence of dental caries, pulpitis, gingivitis, periodontitis, oral ulceration, and stomatitis were calculated using multiple Poisson regression models.

### Results

A significantly higher prevalence (74.6% vs. 63.0%, $P = 0.001$) and frequency (median 5.37 vs. 1.45 per year, $P < 0.001$) dental visits were observed in patients with pSS compared with patients in the comparison cohort. The risk of dental caries (adjusted incidence rate ratio [aIRR] 1.64, $P < 0.001$), pulpitis (aIRR 1.42, $P < 0.001$), gingivitis (aIRR 1.43, $P < 0.001$), periodontitis (aIRR 1.44, $P < 0.001$), oral ulceration (aIRR 1.98, $P < 0.001$), and stomatitis (aIRR 2.06, $P < 0.001$) were significantly higher in patients with pSS.

Requests for data can be sent as a formal application to the Health and Welfare Data Science Center, Department of Statistics, Ministry of Health and Welfare, Taiwan (http://dep.mohw.gov.tw/DOS/np-2497-113.html).

**Funding:** The authors received no specific funding for this work.

**Competing interests:** The authors have declared that no competing interests exist.

## Conclusions

In this nationwide, population-based cohort study, a higher prevalence and frequency of dental visits were found in patients with pSS. Patients with PSS had increased risk of six most common dental disorders, including dental caries, pulpitis, gingivitis, periodontitis, oral ulceration, and stomatitis. Rheumatologists should remain vigilant for the dental health of patients with pSS.

## Introduction

Primary Sjögren's syndrome (pSS) is chronic autoimmune disease characterized by an abnormal immune response to salivary and lacrimal gland resulting in dry mouth and dry eye [1,2]. Decreased salivary gland function can lead to oral dryness, which is a major clinical characteristic of patients with pSS. Saliva is a complex fluid containing various proteins, lipids, electrolytes, immunoglobulin A, hormones, and buffers [3]. The main functions of saliva include balancing the oral microbiota, providing lubrication and protection of the oral mucosa and tongue, facilitating carbohydrate digestion in the mouth, providing acid-base buffering and mineral salts, acting as a physical obstacle, and involving in wound healing mediated by epidermal growth factor (EGF) and vascular endothelial growth factor (VEGF) [4]. Therefore, decrease salivary gland secretion can cause various dental disorders in patients with pSS.

Several studies have shown that patients with pSS have a high prevalence of dental caries, oral ulceration, or poor gingival and periodontal condition [5,6]. However, the overall dental condition and incidence rate ratio (IRR) for the development of common dental disorders in patients with pSS, compared with patients without pSS, remain unknown. Therefore, the aim of this secondary cohort study was to compare the frequency and percentage of dental visits between patients with and without pSS using the Taiwan's National Health Insurance Research Database (NHIRD) [7], which is a nationwide, population-based database containing comprehensive medical service utilization records of over 99% of Taiwan's 23-million population. In addition, the incidence rates for the top six most common diagnoses of dental disorders in patients with and without pSS, as well as their IRRs were calculated.

## Materials and methods

### Identification of the pSS cohort and a comparison cohort

This is a secondary cohort study using claim data from Taiwan's NHIRD. The study protocol was reviewed and approved by the institutional review board of the Dalin Tzu Chi Hospital, Buddhist Tzu Chi Medical Foundation, Taiwan (No. B10104020). The NHIRD files contain de-identified secondary data, the need for informed consent from individual subjects was waived.

Using the 2000–2012 catastrophic illness datafile, a subset of the NHIRD, patients with pSS were identified based on the International Classification of Diseases, Ninth revision, clinical modification (ICD-9-CM) code 710.2. In Taiwan, pSS is classified as a catastrophic illness, and patients with pSS can apply for a certificate from the National Health Insurance Administration (NHIA). The certification is issued to these patients after their medical records, serological, pathological, and imaging reports were approved by the NHIA, according to the European Study Group on Classification Criteria for Sjögren's Syndrome [8,9]. Approved patients are exempted from copayments for health care expenses incurred for the treatment of their

catastrophic illness. In this study, the application date of the catastrophic illness certificate was defined as the index date for the pSS. Patients between the ages of 20 to 80 years on the index date were included in the study.

A comparison cohort was assembled from a random sample of the outpatient datafile of the 2000 Longitudinal Health Insurance Database (LHID 2000), which contains health claims data, from January 1, 2000 to December 31, 2012, for one million beneficiaries randomly sampled from all health insurance enrollees of the NHIRD in 2000. In the present study, 10 patients were selected, based on frequency matching for sex, 5-year age interval, and index year, for each patient with pSS to form comparison cohort.

A number of diseases were identified and evaluated as potential confounding variables in this study, and they included hypertension (ICD-9-CM codes: 401, 402, 403, 404, and 405), diabetes mellitus (ICD-9-CM codes: 250.0, 250.1, 250.2, 250.3, 250.4, 250.5, 250.6, and 250.7), congestive heart failure (ICD-9-CM code: 428), chronic pulmonary disease (ICD-9-CM codes: 49x, 500−505, and 506.4), malignancy (ICD-9-CM codes: 14x, 15x, 16x, 170−179, 18x, 190 −198, 200−203, 204−208, 199.1, and 199.0), dyslipidemia (ICD-9-CM code: 272), coronary artery disease (ICD-9-CM codes: 414.01 and 414.00), myocardial infraction (ICD-9-CM codes: 410 and 412), peripheral vascular disease (ICD-9-CM codes: 443.9, 441 and 785.4), cerebrovascular disease (ICD-9-CM code: 43x), dementia (ICD-9-CM code: 290), musculoskeletal disorders (ICD-9-CM codes: 710 except 710.0, 714, 717, 720, and 725), peptic ulcer disease (ICD-9-CM codes: 531−534), chronic kidney disease (ICD-9-CM codes: 586, 585, 588.8, and 588.9), and liver disease (ICD-9-CM codes: 571.2, 571.4, 571.5, 571.6, 456.0, 456.1, 456.2 and 572). These diseases were defined by three outpatient visits within 90 days that occurred within one year prior to the index date.

### Identification of dental diseases

Both the pSS cohort and the comparison cohort were followed until our study events had occurred or when the end of the follow-up period had reached. We collected data of dental visits up to 90 days after the index date. First, we reviewed all the dental diseases of patients with pSS using ICD-9-CM codes and selected the top six most common dental diseases based on the frequency of dental visits for further analysis. The dental diseases evaluated included (1) dental caries (ICD-9 code: 521.0), (2) pulpitis (ICD-9 code: 522.0), (3) gingivitis (ICD-9 codes: 523.0, 523.1, and 523.2), (4) periodontitis (ICD-9 codes: 523.3, 523.4, 523.5, and 523.8), (5) oral ulceration (ICD-9 code: 528.2) and (6) stomatitis (ICD-9 code: 528.0). The incidence rates and incidence rate ratios of each of these six outcomes of interest in patients with pSS and controls were calculated. The follow-up duration used for the calculation of person-years was obtained by subtracting the date when the outcome of interest occurred with the index date. Patients who were diagnosed with the outcome of interest within 90 days before the index date were excluded from the study.

### Statistical analysis

The basic characteristics between the pSS cohort and the comparison cohort were compared using Chi-square test, t-test, or non-parametric Mann-Whitney U-test, as appropriate. The prevalence and frequency of the overall dental diseases and each of the six specific dental diseases between the pSS cohort and the comparison cohort were compared using Chi-square test and Mann-Whitney U-test, respectively. As the data for the number of visits were not normally distributed, they are presented as median and interquartile range, in addition to mean and standard deviation.

The incidence rate per 1,000 person-years was calculated for the pSS cohort and the comparison cohort. IRRs for the outcome variables were calculated using Poisson regression models (i.e., generalized linear models with a Poisson log-linear link function and person-years as the offset variable), with and without adjusting for the potential confounding variables, including age, sex, socioeconomic status, geographical region, and comorbidities. Additional subgroup analyses were also performed with stratification by sex, and age groups (20–55 and $> 55$ years). A two-tailed $P$ value of $< 0.05$ was considered statistically significant. All statistical analyses were conducted using IBM SPSS Statistics for Windows, version 24.0 (IBM Corp, Armonk, NY, USA).

## Results

### Basic characteristics and comorbidities of patients in the pSS cohort and the comparison cohort

The basic characteristics and comorbidities of the 709 patients in the pSS cohort and 7090 patients without PSS in the comparison cohort were shown in Table 1. There were no significant differences between the two cohorts with respect to sex and age. The socioeconomic status as estimated by insurance premium level and the geographic region were significantly higher in the pSS cohort.

We also compared the prevalence of comorbidities between the pSS and the comparison cohort. Patients with pSS showed a significantly higher proportion of peripheral vascular diseases, peptic ulcer diseases, musculoskeletal diseases, and liver diseases, but a significantly lower proportion of diabetes mellitus, compared with those in the comparison cohort.

### Frequency, proportion, and common causes of dental visits

There was a higher prevalence (74.6% vs. 63.0%, $P < 0.001$) and frequency (medium 5.37 vs. 1.45 per year, $P < 0.001$) of dental visits in patients with pSS compared with those without (Table 2). We analyzed the causes of dental visit in patients with pSS and selected the six most common dental diseases for further analysis. Table 2 shows the prevalence and frequency of six top most common dental diseases including dental caries, pulpitis, gingivitis, periodontitis, oral ulceration, and stomatitis. The prevalence and frequency of dental visits for all six dental diseases were significantly higher in the pSS cohort compared with the comparison cohort.

### Risk of developing dental caries and pulpitis in patients with pSS

Table 3 showed the incidence rates and IRRs for developing dental caries and its complication, pulpitis, in the pSS cohort and the comparison cohort, with and without stratification by sex and age group. Overall, the pSS cohort showed a significantly higher incidence rate of developing dental caries (adjusted IRR 1.64, 95% CI: 1.47–1.84, $P < 0.001$) compared with the comparison cohort. Both male and female patients with pSS had a significantly higher risk of developing dental caries (adjusted IRR 1.43, 95% CI: 1.00–2.03, $P = 0.047$ and adjusted IRR 1.67, 95% CI: 1.48–1.88, $P < 0.001$, respectively). In addition, the risk of developing dental caries in pSS patients was significantly higher in both the 20–55 years age group (adjusted IRR 1.84, CI: 1.59–2.13; $P < 0.001$) and $> 55$ years age group (adjusted IRR 1.43, CI: 1.19–1.72, $P < 0.001$) compared with those in the comparison cohort. The risk of developing dental caries in pSS patients was significantly higher with a diseases duration of $\leq 5$ years (adjusted IRR 1.50, CI: 1.33–1.68; $P < 0.001$) when compared with the comparison cohort. However, it was not significantly higher in those with a disease duration of $> 5$ years (adjusted IRR 1.28, CI: 0.80–2.06, $P < 0.001$) when compared with the comparison cohort.

**Table 1. Basic characteristics of the Sjögren's syndrome cohort and the comparison cohort (N = 7799).**

| Variable | n (%) | | | | P |
|---|---|---|---|---|---|
| | Sjogren's syndrome cohort 709 (9.1) | | comparison cohort 7090 (90.9) | | |
| Sex | | | | | > 0.999 |
| Male | 79 | (11.1) | 790 | (11.1) | |
| Female | 6300 | (88.9) | 630 | (88.9) | |
| Age interval at entry, years | | | | | > 0.999 |
| 20−54 | 384 | (54.2) | 3840 | (54.2) | |
| > 55 | 325 | (45.8) | 3250 | (45.8) | |
| Mean age (standard deviation), years | 53.1 | (13.6) | 53.1 | (13.6) | > 0.999 |
| Median (interquartile range), years | 53.0 | (44−63) | 53.0 | (44−63) | |
| Salary grades for insurance (n = 7789) | | | | | < 0.001 |
| ≤ 19,000 | 327 | (46.3) | 3786 | (53.5) | |
| 19,001−24,000 | 235 | (33.3) | 2208 | (31.2) | |
| ≥ 24,001 | 144 | (20.4) | 1089 | (15.4) | |
| Geographic region (n = 7534) | | | | | < 0.001 |
| Northern | 367 | (54.0) | 4223 | (61.6) | |
| Central | 176 | (25.9) | 1061 | (15.5) | |
| Southern | 129 | (19.0) | 1423 | (20.8) | |
| Eastern | 8 | (1.1) | 147 | (2.1) | |
| Comorbidities | | | | | |
| Hypertension | 62 | (8.7) | 547 | (7.6) | 0.330 |
| Diabetes | 15 | (2.1) | 340 | (4.8) | 0.001 |
| Congestive heart failure | 3 | (0.4) | 35 | (0.5) | 0.797 |
| Chronic pulmonary disease | 17 | (2.4) | 147 | (2.1) | 0.566 |
| Cancer | 14 | (2.0) | 125 | (1.8) | 0.685 |
| Dyslipidemia | 25 | (3.5) | 220 | (3.1) | 0.538 |
| Coronary artery disease | 4 | (0.6) | 24 | (0.3) | 0.338 |
| Prior myocardial infarction | 0 | (0.0) | 6 | (0.1) | 0.438 |
| Peripheral vascular disease | 3 | (0.4) | 4 | (0.1) | 0.002 |
| Cerebrovascular disease | 8 | (1.1) | 106 | (1.5) | 0.438 |
| Dementia | 1 | (0.1) | 21 | (0.3) | 0.458 |
| Musculoskeletal diseases | 48 | (6.8) | 43 | (0.6) | < 0.001 |
| Peptic ulcer disease | 31 | (4.4) | 181 | (2.6) | 0.005 |
| Chronic kidney disease | 9 | (1.3) | 72 | (1.0) | 0.525 |
| Liver disease | 20 | (2.8) | 88 | (1.2) | 0.001 |

Socioeconomic status was estimated by insurance premiums based on salary. Low: < 19,000 New Taiwan dollars (NT$); middle: 19,001−24,000; and high: > 24,000.

P values were obtained by Chi-square test for categorical variables and t-test or Mann-Whitney U-test for continuous variables, as appropriate.

For the development of pulpitis, the pSS cohort showed a significantly higher incidence rate of developing pulpitis (adjusted IRR 1.42, 95% CI: 1.22−1.64, P < 0.001) compared with the comparison cohort. In the stratified analysis, only female patients with pSS had a significantly increased risk (adjusted IRR 1.51, 95% CI: 1.29−1.76, P < 0.001). The risk of developing pulpitis in pSS patients was significantly elevated only in the 20−55 years age group (adjusted IRR 1.70, 95% CI: 1.41−2.04, P < 0.001). The risk of developing pulpitis was significantly higher in pSS patients compared with the comparison cohort, regardless of whether their disease duration was ≤ 5 years or > 5 years (adjusted IRR 1.37, 95% CI: 1.17−1.62, P < 0.001 and adjusted IRR 1.62, 95% CI: 1.13−2.34, P = 0.009, respectively).

**Table 2. The prevalence and frequency of dental disorders in the Sjögren's syndrome cohort and the comparison cohort (N = 7799).**

| Variable | n (%) | | | | *P* |
|---|---|---|---|---|---|
| | **Sjögren's syndrome cohort 709 (9.1)** | | **Comparison cohort 7090 (90.9)** | | |
| **Dental visits** | | | | | |
| Prevalence (%) | 529 | (74.6) | 4465 | (63.0) | < 0.001 |
| Number of visits, median (IQR) (per year) | 5.37 | (0.00–20.89) | 1.45 | (0.00–9.10) | < 0.001 |
| Number of visits, mean (SD) (per year) | 19.53 | (37.88) | 9.50 | (20.68) | |
| **Dental caries** (ICD-9-CM 521.0) | | | | | |
| Prevalence (%) | 443 | (62.5) | 3563 | (50.3) | < 0.001 |
| Number of visits, median (IQR) (per year) | 1.71 | (0.00–8.25) | 0.15 | (0.00–3.41) | < 0.001 |
| Number of visits, mean (SD) (per year) | 8.16 | (18.99) | 3.76 | (8.60) | |
| **Periodontitis** (ICD-9-CM 523.3, 523.4, 523.5, 523.8) | | | | | |
| Prevalence (%) | 440 | (62.1) | 3581 | (50.5) | < 0.001 |
| Number of visits, median (IQR) (per year) | 1.25 | (0.00–6.49) | 0.15 | (0.00–3.11) | < 0.001 |
| Number of visits, mean (SD) (per year) | 6.68 | (14.22) | 3.59 | (9.29) | |
| **Pulpitis** (ICD-9-CM 522.0) | | | | | |
| Prevalence (%) | 241 | (34.0) | 1776 | (25.0) | < 0.001 |
| Number of visits, median (IQR) (per year) | 0.00 | (0.00–1.33) | 0.00 | (0.00–0.10) | < 0.001 |
| Number of visits, mean (SD) (per year) | 2.26 | (6.90) | 1.05 | (3.24) | |
| **Gingivitis** (ICD-9-CM 523.0, 523.1, 523.2) | | | | | |
| Prevalence (%) | 263 | (37.1) | 2049 | (28.9) | < 0.001 |
| Number of visits, median (IQR) (per year) | 0.00 | (0.00–1.58) | 0.00 | (0.00–0.52) | < 0.001 |
| Number of visits, mean (SD) (per year) | 1.94 | (4.84) | 1.17 | (3.89) | |
| **Oral ulceration** (ICD-9-CM 528.2) | | | | | |
| Prevalence (%) | 166 | (23.4) | 890 | (12.6) | < 0.001 |
| Number of visits, median (IQR) (per year) | 0.00 | (0.00–0.00) | 0.00 | (0.00–0.00) | < 0.001 |
| Number of visits, mean (SD) (per year) | 1.25 | (4.40) | 0.48 | (2.46) | |
| **Stomatitis** (ICD-9-CM 528.0) | | | | | |
| Prevalence (%) | 113 | (15.9) | 483 | (6.8) | < 0.001 |
| Number of visits, median (IQR) (per year) | 0.00 | (0.00–0.00) | 0.00 | (0.00–0.00) | < 0.001 |
| Number of visits, mean (SD) (per year) | 0.76 | (2.76) | 0.25 | (1.88) | |

ICD-9-CM: International Classification of Diseases, Ninth revision, clinical modification; IQR: Interquartile range; SD: Standard deviation.

*P* values were obtained by Chi-square test for comparison of prevalence and Mann-Whitney U-test for comparison of medians of number of visits.

## Risk of developing gingivitis and periodontitis in patients with pSS

Table 4 shows the incidence rates and IRRs for developing gingivitis and its severe form, peri-odontitis, in the pSS cohort and the comparison cohort, with and without stratification by sex and age group. Patients in the pSS cohort had a significantly higher incidence of developing gingivitis compared with the comparison cohort (adjusted IRR 1.43, 95% CI: 1.24−1.65, $P < 0.001$). Only female patients with pSS had a higher risk of developing gingivitis (adjusted IRR 1.44, 95% CI: 1.24−1.67, $P < 0.001$). In addition, the 20−55 years age group patients with pSS showed a significant higher risk of developing gingivitis (adjusted IRR 1.56, 95% CI: 1.31−1.87, $P < 0.001$). The risk of developing gingivitis was significantly higher in pSS patients compared with comparison cohort, regardless of whether their disease duration was ≤ 5 years or > 5 years (adjusted IRR 1.39, 95% CI: 1.19−1.63, $P < 0.001$ and adjusted IRR 1.47, 95% CI: 1.04−2.07, $P = 0.028$, respectively).

For the development of periodontitis, patients in the pSS cohort had a significantly higher incidence of developing periodontitis compared with the comparison cohort (adjusted IRR

**Table 3. Incidence rates and incidence risk ratios of dental caries and pulpitis in the Sjögren's syndrome cohort and the comparison cohort.**

| Disorder (ICD-9-CM) | Group | Sjögren's syndrome cohort | | | Comparison cohort | | | IRR (95% CI) | Adjusted IRR (95% CI) |
|---|---|---|---|---|---|---|---|---|---|
| | | No. of patient | Person-years | IR | No. of patient | Person-years | IR | *P* | *P* |
| **Dental caries** (523.X) | | | | | | | | | |
| | **Overall** | 358 | 1243 | 288.01 | 3135 | 17400 | 180.17 | 1.60 (1.43−1.78) < 0.001 | 1.64 (1.47−1.84) < 0.001 |
| | **Sex** | | | | | | | | |
| | male | 38 | 164 | 231.71 | 349 | 2019 | 172.86 | 1.34 (0.96−1.87) 0.086 | 1.43 (1.00−2.03) 0.047 |
| | female | 320 | 1079 | 296.57 | 2786 | 15381 | 181.13 | 1.64 (1.46−1.84) < 0.001 | 1.67 (1.48−1.88) < 0.001 |
| | **Age group, years** | | | | | | | | |
| | 20−55 | 222 | 579 | 222 | 1911 | 9168 | 208.44 | 1.84 (1.60−2.11) < 0.001 | 1.84 (1.59−2.13) < 0.001 |
| | > 55 | 136 | 664 | 136 | 1224 | 8232 | 148.69 | 1.38 (1.16−1.64) < 0.001 | 1.43 (1.19−1.72) < 0.001 |
| | **Disease duration, years** | | | | | | | | |
| | ≤ 5 | 338 | 737 | 458.62 | 2861 | 9531 | 300.18 | 1.53 (1.36−1.71) < 0.001 | 1.50 (1.33−1.68) < 0.001 |
| | > 5 | 20 | 506 | 39.53 | 274 | 7869 | 34.82 | 1.14 (0.72−1.79) 0.583 | 1.28 (0.80−2.06) 0.312 |
| **Pulpitis** (522.0) | | | | | | | | | |
| | **Overall** | 220 | 2256 | 97.52 | 1694 | 24981 | 67.81 | 1.44 (1.25−1.66) < 0.001 | 1.42 (1.22−1.64) < 0.001 |
| | **Sex** | | | | | | | | |
| | Male | 18 | 302 | 59.60 | 193 | 2794 | 69.08 | 0.86 (0.53−1.40) 0.551 | 0.86 (0.51−1.43) 0.554 |
| | female | 202 | 1954 | 103.38 | 1501 | 22187 | 67.65 | 1.53 (1.32−1.77) < 0.001 | 1.51 (1.29−1.76) < 0.001 |
| | **Age group, years** | | | | | | | | |
| | 20−55 | 143 | 1214 | 117.79 | 1035 | 14262 | 72.57 | 1.62 (1.36−1.93) < 0.001 | 1.70 (1.41−2.04) < 0.001 |
| | > 55 | 77 | 1042 | 73.90 | 659 | 10719 | 61.48 | 1.20 (0.95−1.52) 0.127 | 1.10 (0.86−1.42) 0.443 |
| | **Disease duration, years** | | | | | | | | |
| | ≤ 5 | 184 | 956 | 192.47 | 1433 | 10689 | 134.06 | 1.44 (1.23−1.67) < 0.001 | 1.37 (1.17−1.62) < 0.001 |
| | > 5 | 36 | 1300 | 27.69 | 261 | 14292 | 18.26 | 1.52 (1.07−2.15) 0.019 | 1.62 (1.13−2.34) 0.009 |

CI: Confidence interval; ICD-9-CM: International Classification of Diseases, Ninth revision, clinical modification; IR: Incidence rate per 1,000 person-years; IRR: Incidence rate ratio.

aIRR adjusted for age, sex, socioeconomic status, geographic region, hypertension, diabetes, congestive heart failure, chronic pulmonary disease, cancer, dyslipidemia, coronary artery disease, prior myocardial infarction, peripheral vascular disease, cerebrovascular disease, dementia, rheumatologic disease, peptic ulcer disease, chronic kidney disease, and liver disease.

1.44, 95% CI: 1.28−1.62, $P < 0.001$). Both male and female patients with pSS showed a significantly increased risk of developing periodontitis (adjusted IRR 1.46, 95% CI: 1.02−2.09, $P = 0.038$ and adjusted IRR 1.43, 95% CI: 1.26−1.62, $P < 0.001$, respectively). In addition,

**Table 4. Incidence rates and incidence risk ratios of gingivitis and periodontitis in the Sjögren's syndrome cohort and the comparison cohort.**

| Disorder (ICD-9-CM) | Group | Sjögren's syndrome cohort | | | Comparison cohort | | | IRR (95% CI) | Adjusted IRR (95% CI) |
|---|---|---|---|---|---|---|---|---|---|
| | | No. of patient | Person-years | IR | No. of patient | Person-years | IR | *P* | *P* |
| **Gingivitis** (523.0x, 523.1x, 523.2x) | | | | | | | | | |
| | **Overall** | 237 | 2262 | 104.77 | 1887 | 24980 | 75.54 | 1.39 (1.21–1.59) < 0.001 | 1.43 (1.24–1.65) < 0.001 |
| | **Sex** | | | | | | | | |
| | male | 25 | 283 | 88.34 | 204 | 2824 | 72.24 | 1.22 (0.81–1.85) 0.344 | 1.29 (0.82–2.02) 0.269 |
| | female | 212 | 1979 | 107.12 | 1683 | 22157 | 75.96 | 1.41 (1.22–1.63) < 0.001 | 1.44 (1.24–1.67) < 0.001 |
| | **Age group, years** | | | | | | | | |
| | 20–55 | 155 | 1175 | 131.91 | 1197 | 13982 | 85.61 | 1.54 (1.30–1.82) < 0.001 | 1.56 (1.31–1.87) < 0.001 |
| | > 55 | 82 | 1087 | 75.44 | 690 | 10998 | 62.74 | 1.20 (0.96–1.52) 0.114 | 1.22 (0.96–1.55) 0.101 |
| | **Disease duration, years** | | | | | | | | |
| | ≤ 5 | 195 | 922 | 211.50 | 1555 | 10298 | 151.00 | 1.40 (1.21–1.63) < 0.001 | 1.39 (1.19–1.63) < 0.001 |
| | > 5 | 42 | 1340 | 31.34 | 332 | 14682 | 22.61 | 1.39 (1.01–1.91) 0.046 | 1.47 (1.04–2.07) 0.028 |
| **Periodontitis** (523.3x, 523.4x, 523.5x, 523.8x) | | | | | | | | | |
| | **Overall** | 339 | 1308 | 259.17 | 3084 | 17090 | 180.46 | 1.44 (1.28–1.61) < 0.001 | 1.44 (1.28–1.62) < 0.001 |
| | **Sex** | | | | | | | | |
| | male | 37 | 145 | 255.17 | 364 | 1890 | 192.59 | 1.32 (0.94–1.85) 0.106 | 1.46 (1.02–2.09) 0.038 |
| | female | 302 | 1163 | 259.67 | 2720 | 15200 | 178.95 | 1.45 (1.29–1.63) < 0.001 | 1.43 (1.26–1.62) < 0.001 |
| | **Age group, years** | | | | | | | | |
| | 20–55 | 198 | 719 | 275.38 | 1841 | 9344 | 197.02 | 1.40 (1.21–1.62) < 0.001 | 1.40 (1.20–1.63) < 0.001 |
| | > 55 | 141 | 589 | 239.39 | 1243 | 7746 | 160.47 | 1.49 (1.25–1.78) < 0.001 | 1.52 (1.27–1.82) < 0.001 |
| | **Disease duration, years** | | | | | | | | |
| | ≤ 5 | 308 | 761 | 404.73 | 2812 | 9297 | 302.46 | 1.34 (1.19–1.50) < 0.001 | 1.29 (1.14–1.46) < 0.001 |
| | > 5 | 31 | 547 | 56.67 | 272 | 7793 | 34.90 | 1.62 (1.12–2.35) 0.011 | 1.75 (1.18–2.57) 0.005 |

CI: Confidence interval; ICD-9-CM: International Classification of Diseases, Ninth revision, clinical modification; IR: Incidence rate per 1,000 person-years; IRR: Incidence rate ratio.

Adjusted for age, sex, socioeconomic status, geographic region, hypertension, diabetes, congestive heart failure, chronic pulmonary disease, cancer, dyslipidemia, coronary artery disease, prior myocardial infarction, peripheral vascular disease, cerebrovascular disease, dementia, rheumatologic disease, peptic ulcer disease, chronic kidney disease, and liver disease.

patients with pSS in both age groups showed a significant higher risk of developing periodontitis (adjusted IRR 1.40, 95% CI: 1.20−1.63, *P* < 0.001 and adjusted IRR 1.52, 95% CI: 1.27−1.82, *P* < 0.001, respectively). The risk of developing periodontitis in pSS patients was significantly

higher compared with comparison cohort, regardless of whether their diseases duration was ≤ 5 years or > 5 years (adjusted IRR 1.29, 95% CI: 1.14–1.46, $P < 0.001$ and adjusted IRR 1.75, 95% CI: 1.18–2.57, $P = 0.005$, respectively).

### Risk of developing oral ulceration and stomatitis in patients with pSS

Table 5 shows the incidence rates and IRRs for developing oral ulceration and its related condition, stomatitis, in the pSS cohort and the comparison cohort, with and without stratification by sex and age group. Patients in the pSS cohort exhibited a significantly higher incidence of developing oral ulceration compared with those in the comparison cohort (adjusted IRR 1.98, 95% CI: 1.65–2.38, $P < 0.001$). Only female patients with pSS had a significantly increased risk of developing oral ulceration (adjusted IRR 2.04, 95% CI: 1.68–2.46, $P < 0.001$). In addition, patients with pSS in both age groups showed a significant higher risk of developing oral ulceration (adjusted IRR 2.11, 95% CI: 1.64–2.71, $P < 0.001$ and adjusted IRR 1.87, 95% CI: 1.44–2.44, $P < 0.001$, respectively). The risk of developing oral ulceration in pSS patients was significantly higher compared with the comparison cohort, regardless of whether their disease duration was ≤ 5 years or > 5 years (adjusted IRR 1.91, 95% CI: 1.56–2.34, $P < 0.001$ and adjusted IRR 1.62, 95% CI: 1.09–2.40, $P = 0.016$, respectively).

For the development of stomatitis, patients in the pSS cohort also exhibited a significantly higher incidence of developing stomatitis compared with those in the comparison cohort (adjusted IRR 2.06, 95% CI: 1.64–2.61, $P < 0.001$, respectively). Only female patients with pSS had a significantly increased risk of developing oral ulceration (adjusted IRR 2.23; 95% CI: 1.74–2.84, $P < 0.001$). In addition, patients with pSS in both age groups showed a significant higher risk of developing stomatitis (adjusted IRR 1.90, 95% CI: 1.37–2.66, $P < 0.001$ and adjusted IRR 2.29, 95% CI: 1.65–3.17, $P < 0.001$, respectively). The risk of developing stomatitis in pSS patients was significantly higher compared with the comparison cohort, regardless of whether their disease duration was ≤ 5 years or > 5 years (adjusted IRR 1.97, 95% CI: 1.52–2.56, $P < 0.001$ and adjusted IRR 1.86, 95% CI: 1.06–3.23, $P = 0.029$, respectively).

### Discussion

Although it is well known that patients with pSS have impaired dental conditions [10], findings from this study confirmed that these patients had not only an increased risk of developing various common dental diseases, including dental caries, pulpitis, gingivitis, periodontitis, oral ulceration, and stomatitis compared with patients with pSS, but also provided overall as well as sex- and age group-stratified incidence rates and incidence rate ratios. Patients with pSS showed an increased risk of developing dental caries regardless of sex and age group, whereas, its complication, pulpitis, only female and younger patients showed an increased risk. This finding is consistent with previous studies [11,12]. The development of dental caries and its complication, pulpitis can be attributed to the decreased saliva flow and changes in saliva component resulting in a dramatic change of the oral microflora [13]. On the other hands, there are conflicting reports regarding whether patients with pSS have an increased risk of developing periodontitis [6,14,15]. Our findings supported that patients with pSS had an increased risk of developing periodontitis regardless of sex and age group. For gingivitis, only female and younger patients with pSS showed an increased risk compared with patients without pSS. The pathogenesis and mechanisms of periodontitis and pSS are highly complex, and free fatty acids, especially palmitic acid might play a role [16]. Since decreased saliva can impair its lubrication function, the risk of developing stomatitis and oral ulceration was expected to increase. However, to the best of our knowledge, no studies have investigated this issue. Our study

**Table 5. Incidence rates and incidence risk ratios of oral ulceration and stomatitis in the Sjögren's syndrome cohort and the comparison cohort.**

| Disorder (ICD-9-CM) | Group | Sjögren's syndrome cohort | | | Comparison cohort | | | IRR (95% CI) | Adjusted IRR (95% CI) |
|---|---|---|---|---|---|---|---|---|---|
| | | No. of patient | Person-years | IR | No. of patient | Person-years | IR | *P* | *P* |
| **Oral ulceration (528.2)** | | | | | | | | | |
| | **Overall** | 157 | 2608 | 60.20 | 855 | 29655 | 28.83 | 2.09 (1.76–2.48) < 0.001 | 1.98 (1.65–2.38) < 0.001 |
| | **Sex** | | | | | | | | |
| | Male | 16 | 308 | 51.95 | 95 | 3321 | 28.61 | 1.82 (1.07–3.08) 0.027 | 1.51 (0.85–2.69) 0.155 |
| | Female | 141 | 2300 | 61.30 | 760 | 26333 | 28.86 | 2.12 (1.77–2.54) < 0.001 | 2.04 (1.68–2.46) < 0.001 |
| | **Age group, years** | | | | | | | | |
| | 20–55 | 85 | 1492 | 56.97 | 442 | 17258 | 25.61 | 2.22 (1.76–2.81) < 0.001 | 2.11 (1.64–2.71) < 0.001 |
| | > 55 | 72 | 1117 | 64.46 | 413 | 12397 | 33.31 | 1.93 (1.51–2.49) < 0.001 | 1.87 (1.44–2.44) < 0.001 |
| | **Disease duration, years** | | | | | | | | |
| | ≤ 5 | 125 | 1018 | 122.79 | 635 | 10471 | 60.64 | 2.02 (1.67–2.45) < 0.001 | 1.91 (1.56–2.34) < 0.001 |
| | > 5 | 32 | 1590 | 20.13 | 220 | 19183 | 11.47 | 1.76 (1.21–2.54) 0.003 | 1.62 (1.09–2.40) 0.016 |
| **Stomatitis (528.0)** | | | | | | | | | |
| | **Overall** | 100 | 2730 | 36.63 | 471 | 30722 | 15.33 | 2.39 (1.93–2.96) < 0.001 | 2.06 (1.64–2.61) < 0.001 |
| | **Sex** | | | | | | | | |
| | male | 9 | 332 | 27.11 | 65 | 3409 | 19.07 | 1.42 (0.71–2.86) 0.321 | 1.02 (0.45–2.32) 0.959 |
| | female | 91 | 2398 | 37.95 | 406 | 27313 | 14.86 | 2.55 (2.03–3.20) < 0.001 | 2.23 (1.74–2.84) < 0.001 |
| | **Age group, years** | | | | | | | | |
| | 20–55 | 49 | 1598 | 30.66 | 250 | 17916 | 13.95 | 2.20 (1.62–2.98) < 0.001 | 1.90 (1.37–2.66) < 0.001 |
| | > 55 | 51 | 1132 | 45.05 | 221 | 12806 | 17.26 | 2.61 (1.93–3.54) < 0.001 | 2.29 (1.65–3.17) < 0.001 |
| | **Disease duration, years** | | | | | | | | |
| | ≤ 5 | 82 | 985 | 83.25 | 374 | 10555 | 35.43 | 2.35 (1.85–2.98) < 0.001 | 1.97 (1.52–2.56) < 0.001 |
| | > 5 | 18 | 1745 | 10.32 | 97 | 20166 | 4.81 | 2.14 (1.30–3.55) 0.003 | 1.86 (1.06–3.23) 0.029 |

CI: Confidence interval; ICD-9-CM: International Classification of Diseases, Ninth revision, clinical modification; IR: Incidence rate per 1,000 person-years; IRR: Incidence rate ratio.

Adjusted for age, sex, socioeconomic status, geographic region, hypertension, diabetes, congestive heart failure, chronic pulmonary disease, cancer, dyslipidemia, coronary artery disease, prior myocardial infarction, peripheral vascular disease, cerebrovascular disease, dementia, rheumatologic disease, peptic ulcer disease, chronic kidney disease, and liver disease.

showed that the risk of developing stomatitis and oral ulceration was significantly higher in female patients with pSS, and both in younger and older age groups.

An increased risk of developing common dental diseases, including dental caries, pulpitis, gingivitis, periodontitis, oral ulceration, and stomatitis in patients with pSS could lead to

difficulty in speaking, chewing, and swallowing. These dental diseases can eventually lead to early teeth loss in patients with pSS, resulting in the need for removable prostheses. However, hyposalivation in patient with pSS can affect the retention of removable prostheses because saliva is necessary to create adhesion, cohesion, and surface tension for denture retention [17]. As a result, proper digestive and nutrient extraction functions may be impaired in these patients. In addition, chronic periodontitis might be associated with an increase in the risk of developing a number of systemic diseases, including cardiovascular disease, gastrointestinal diseases, colorectal cancer, diabetes, insulin resistance, Alzheimer's disease, and rheumatoid arthritis [18,19]. Therefore, it is crucial to prevent the development dental diseases and their sequel. Currently, muscarinic agonists (pilocarpine hydrochloride and cevimeline hydrochloride) are commonly used to increase saliva section for controlling oral dryness [2], but the protective effect of these agents on dental caries and periodontitis was still equivocal [20]. Topical fluoride can effectively decrease dental caries in patients with pSS [21]. Regular dental visits to receive professional dental cleaning and oral hygiene instructions should be encouraged in patients with pSS [22] to lower the incidence of dental diseases in these patients.

A few limitations of this study should be mentioned. First, the sample size for male patients with pSS was relatively small (n = 79), which might be insufficient to show the differences in IRRs between male patients with and without pSS. Second, the analyses relied on the claim-based NHIRD and therefore, those who did not use dental services were not included. Third, detailed serology data and clinical severity of dry mouth were lacking in patients with pSS, and therefore these variables could not be adjusted in our analyses. It is known that the salivary flow is significantly reduced with a longer duration of pSS [23]. Nevertheless, we found that the IRRs for pulpitis, gingivitis, periodontitis, oral ulceration, and stomatitis were similar between a disease duration of $\leq$ 5 years and $>$ 5 years in patients with pSS compared with those in the comparison cohort. In fact, the IRR for of dental caries was not significantly elevated in pSS patients with a disease duration longer than 5 years. The development of these dental diseases is highly complex. Although xerostomia may be an important factor, other physical, biological, environmental, behavioral, and lifestyle factors can also play critical roles in the pathogenesis of different dental diseases [24].

In conclusion, we found that patients with pSS showed a higher risk for developing various common dental diseases, including dental caries, pulpitis, gingivitis, periodontitis, oral ulceration, and stomatitis. Rheumatologists should remain vigilant for the dental health of patients with pSS.

## Acknowledgments

This study is based in part on data from the National Health Insurance Research Database provided by the National Health Insurance Administration, Ministry of Health and Welfare and managed by the National Health Research Institutes, Taiwan. The interpretation and conclusions contained herein do not represent those of the National Health Insurance Administration, Ministry of Health and Welfare or the National Health Research Institutes, Taiwan.

## Author Contributions

**Conceptualization:** Chi-Jou Chuang, Ming-Chi Lu, Malcolm Koo.

**Formal analysis:** Chia-Wen Hsu.

**Writing – original draft:** Chi-Jou Chuang, Ming-Chi Lu.

**Writing – review & editing:** Malcolm Koo.

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
