## [Decision Letter · Decision Letter 0]

24 Aug 2020

PONE-D-20-05093

Increased risk of developing dental diseases in patients with primary Sjögren’s syndrome– a secondary cohort analysis of population-based claims data

PLOS ONE

Dear Dr. Koo,

Thank you for submitting your manuscript to PLOS ONE. After careful consideration, we feel that it has merit but does not fully meet PLOS ONE’s publication criteria as it currently stands. Therefore, we invite you to submit a revised version of the manuscript that addresses the points raised during the review process.

We look forward to receiving your revised manuscript.

Kind regards,

Frédéric Denis, Ph.D.

Academic Editor

PLOS ONE

Journal Requirements:

2. We noticed you have some minor occurrence(s) of overlapping text with the following previous publication(s), which needs to be addressed:

https://doi.org/10.1007/s10067-019-04705-z

In your revision ensure you cite all your sources (including your own works), and quote or rephrase any duplicated text outside the Methods section. Further consideration is dependent on these concerns being addressed.

Reviewers' comments:

Reviewer's Responses to Questions

**Comments to the Author**

1. Is the manuscript technically sound, and do the data support the conclusions?

Reviewer #1: Partly

Reviewer #2: Yes

2. Has the statistical analysis been performed appropriately and rigorously? 

Reviewer #1: N/A

Reviewer #2: No

3. Have the authors made all data underlying the findings in their manuscript fully available?

Reviewer #1: Yes

Reviewer #2: Yes

4. Is the manuscript presented in an intelligible fashion and written in standard English?

Reviewer #1: Yes

Reviewer #2: Yes

5. Review Comments to the Author

Reviewer #1: As the aim of your study is to objectify side effects of pSS on oral health, you need to describe more precisely the salivary role : -buffer power

-physical obstacle

-hydration, lubrication

-Monitoring of oral microbiota,

-Tank of ions and minérals salts,

-Healing with EGF and VEGF.

You also need to emphasize on the prevention's problematic as it's difficult to provide a good oral restauration with removable denture when there is a lack of saliva.

By the end, as clinical expression of xerostomia should be different between two patients who suffer of pSS, there is a missing data, you need to objective the clinical severity of dry mouth.

Reviewer #2: Chuang et al are addressing dental disease in the context of Sjögren’s syndrome.

Major comment: No correction for multiple comparisons has been performed.

Minor comments: As the authors address risk factor of dental disease, they may address specifically the effect of treatment for Sjögren’s syndrome (e.g. did MTX-treated patients had more stomatitis? Does treatment affect the risk at all?). They may further address the effect of disease duration, since diagnosis and the one of extraglandular disease on the risk for dental disease.

6. PLOS authors have the option to publish the peer review history of their article (what does this mean?). If published, this will include your full peer review and any attached files.

Reviewer #1: No

Reviewer #2: No

---

## [Editor Report · Decision Letter 1]

7 Sep 2020

Increased risk of developing dental diseases in patients with primary Sjögren’s syndrome– a secondary cohort analysis of population-based claims data

PONE-D-20-05093R1

Dear Dr. Koo,

We’re pleased to inform you that your manuscript has been judged scientifically suitable for publication and will be formally accepted for publication once it meets all outstanding technical requirements.

Kind regards,

Frédéric Denis, Ph.D.

Academic Editor

PLOS ONE
---

## [Editor Report · Acceptance letter]

10 Sep 2020

PONE-D-20-05093R1 

Increased risk of developing dental diseases in patients with primary Sjögren’s syndrome– a secondary cohort analysis of population-based claims data 

Dear Dr. Koo:

I'm pleased to inform you that your manuscript has been deemed suitable for publication in PLOS ONE. Congratulations! Your manuscript is now with our production department. 

Kind regards, 

on behalf of

Dr. Frédéric Denis 

Academic Editor

PLOS ONE